# Multifaceted Linkages among Eco-Climatic Factors, Plankton Abundance, and Gonadal Maturation of Hilsa Shad, *Tenualosa ilisha*, Populations in Bangladesh

Mobin Hossain Shohan [1,†], Mohammad Abu Baker Siddique [2,†], Balaram Mahalder [1], Mohammad Mahfujul Haque [1], Chayon Goswami [3], Md. Borhan Uddin Ahmed [2], Mohammad Ashraful Alam [4], Md. Abul Bashar [4], Yahia Mahmud [5], Mahamudul Alam Chowdhury [6], Md. Mahmudul Hasan [1] and A. K. Shakur Ahammad [2,*]

[1] Department of Aquaculture, Bangladesh Agricultural University, Mymensingh 2202, Bangladesh; shohan.1706097@bau.edu.bd (M.H.S.); balaram.29198@bau.edu.bd (B.M.); mmhaque.aq@bau.edu.bd (M.M.H.); mahmudul.1506011@bau.edu.bd (M.M.H.)

[2] Department of Fisheries Biology and Genetics, Bangladesh Agricultural University, Mymensingh 2202, Bangladesh; absiddique.29199@bau.edu.bd (M.A.B.S.); mbuasiam@gmail.com (M.B.U.A.)

[3] Department of Biochemistry and Molecular Biology, Bangladesh Agricultural University, Mymensingh 2202, Bangladesh; chayon.goswami@bau.edu.bd

[4] Bangladesh Fisheries Research Institute, Riverine Station, Chandpur 3602, Bangladesh; mohammad.bfri@gmail.com (M.A.A.); mabashar.bfri@gmail.com (M.A.B.)

[5] Bangladesh Fisheries Research Institute, Freshwater Station, Mymensingh 2201, Bangladesh; yahiamahmud@yahoo.com

[6] Institute of Marine Sciences, University of Chittagong, Chittagong 4331, Bangladesh; macpbkbdao@gmail.com

* Correspondence: sahammad09@bau.edu.bd; Tel.: +880-171-959-9249; Fax: +880-91-61510

† These authors have contributed equally to this work and share first authorship.

**Abstract:** An integrated multivariate approach was applied to gain a deeper understanding of the feeding biology of hilsa shad, *Tenualosa ilisha*, collected from six different aquatic habitats across Bangladesh. This approach involved linking climatic factors, ecological factors, plankton abundance in water, reproductive traits, and plankton ingestion data. Climatic data were obtained from the National Oceanic and Atmospheric Administration (NOAA) and Climate Data Online (CDO) databases on a monthly basis. Water quality parameters were observed on-site at various sampling sites. Plankton data from water bodies and hilsa guts were collected monthly from the study areas and analyzed in the laboratory. The results obtained were averaged for each month. The correlation tests, multivariate approaches, cluster analyses, and regression analyses revealed that the gonadosomatic index was primarily influenced by climatic factors, the abundance of ingested gut plankton, and their compositions. The analysis of selectivity indices confirmed that plankton preferentially ingested selective taxa. Thirteen plankton groups were identified in the water column of six different hilsa habitats. The dominant phytoplankton groups were Bacillariophyceae (34–53%), Chlorophyceae (31–50%), Cyanophyceae (4–8%), and Euglenophyceae (1–3%). Additionally, Copepoda, Rotifera, and Cladocera were the most numerous zooplankton groups. Hilsa shad primarily consumed Bacillariophyceae (38–57%), Chlorophyceae (35–53%), and Cyanophyceae (4–6%). However, they also exhibited selective ingestion of higher quantities of Bacillariophyceae and Chlorophyceae to fulfill specific and unique metabolic needs. Cluster analysis revealed the variability of phytoplankton and zooplankton abundance in water and gut in relation to diverse locations. Combining all the datasets, a principal component analysis (PCA) was applied. The first two principal components explained more than 54% of the variability. The first component explained the association between the gonadosomatic index and eco-climatic factors, while the second component extracted the cluster of ingested gut plankton in relation to salinity and pH. Pearson's correlations and linear regression analyses showed that the number of gut plankton had a positive influence on the gonadosomatic index (GSI). Finally, the outcomes from these extensive datasets have provided a better understanding of the selective feeding behavior and the influence of feeding biology on the gonadal maturation of *T. ilisha*. This

understanding is likely to be useful for maintaining and improving the growth and productivity of the existing production systems for this transboundary species.

**Keywords:** climatic factors; selective index; plankton abundance; gut content analysis; gonadosomatic index; hilsa fishery

## 1. Introduction

The hilsa shad (*Tenualosa ilisha*), a member of the Clupeidae family, holds immense commercial importance in the Indo-Pacific region, particularly in Bangladesh. This species is prevalent in various environments, including coastal, estuarine, and riverine settings. Remarkably, the hilsa shad stands as the predominant single-species fishery in Bangladesh, constituting about 12 percent (0.53 million MT) of the nation's overall fishery production [1]. Hilsa are found in the rivers of Bangladesh while they migrate from saltwater to freshwater during the spawning season [2]. As an anadromous fish, it seeks out river systems in nearby sea nations for reproduction and returns to the sea after spawning [2,3]. The juvenile hilsa then stays in freshwater for several months to develop before reaching successful recruitment stage [4]. Hilsa, a transboundary fish species, are also found in waters from the Persian Gulf to the Arabian Sea and seek refuge in the shallow waters of the Bay of Bengal [4,5]. Its primary migratory path is from the Meghna River to the Padma and Jamuna Rivers, while a secondary coastal migratory path through the Pashur River, Madhumati River, and Gorai River has also been observed in Bangladesh [6].

Several studies have investigated the feeding habits and diet of hilsa and found that it is a filter feeder that primarily feeds on plankton [7,8]. According to a previous study, hilsa fry (20–40 mm) consume copepods, diatoms, *Daphnia*, and ostracods, while younger hilsa (up to 100 mm) feed on insects, small crustaceans, and polyzoa [9]. Hilsa feeding intensity was highest from January to March and lowest from June to November, and it was observed that hilsa shad eat on the riverbed and move along the bottom zone [9,10]. Understanding the food sources, feeding habits, and feeding habitats of different fish species is important for their effective management and utilization [4]. These earlier studies were performed for a single habitat. A study on the six major hilsa habitats for investigating hilsa feeding biology and behavior has not yet been performed.

Hilsa's food availability can be altered by climate change and standard water quality disruption [11]. Changes in air temperature, precipitation, and wind action can affect the timing and success of upwelling, organic component mixing, and nutrient uptake in the water, resulting the alteration of seasonal succession of plankton and plankton growth in the hilsa feeding habitat. Water quality parameters like dissolved oxygen, pH, and salinity also impact the abundance and diversity of plankton in the water [6,12]. Phytoplankton and zooplankton provide important nutrients for hilsa, making their abundance and availability crucial for the fish's growth and survival [4].

Food plays a crucial role in gonadal development due to its direct influence on an animal's overall health and energy reserves [13]. Adequate nutrition provides essential nutrients, including proteins, lipids, and vitamins, necessary for hormonal regulation and physiological processes involved in reproductive functions [14,15]. In fish, sufficient food intake ensures optimal energy levels required for gamete production and maturation of gonads [13]. A well-balanced diet supports the synthesis of sex hormones, aids in the growth of reproductive tissues, and ultimately facilitates successful gonadal development and reproduction [16]. Insufficient or imbalanced nutrition can lead to reproductive issues, affecting fertility and spawning, highlighting the pivotal role of food in ensuring healthy gonadal development. These factors influence the gonadal activities of the hilsa shad, with studies showing that ecological conditions and temperature fluctuations have significant impacts on breeding periods and gonadal maturation [6,8,17]. The spawning season of

hilsa shad occurs between August and October, with peak spawning observed during the new moon and full moon [6–8].

Extreme climatic events like floods and cyclones have significant effects on the feeding ecology of hilsa fish in Bangladesh [18]. Sedimentation, changes in salinity, and water contamination during floods lead to the destruction of breeding and nursery grounds, alterations in feeding habitats, and disruptions in migration patterns, posing significant challenges to the overall health of aquatic ecosystems [19]. Cyclones can cause extensive damage to the fish's habitat, leading to a reduction in their population [3]. These disasters can create physical barriers in the waterways, making it hard for the fish to swim, affecting their natural life cycle, and resulting in a decrease in population and long-term ecological effects [3,20]. Waste and pollution from human activities can contaminate the water, decreasing the quality of the fish's habitat and causing health problems [8,20]. Collectively, these anthropogenic activities can disrupt the hilsa fish feeding habitat and spawning grounds in Bangladesh, reducing their population and negatively impacting the ecological health of the aquatic ecosystem [3].

Despite an upward trend in the overall hilsa catch, there has been a concerning decline in the year-on-year growth rate of hilsa production in recent years [21]. This decline is attributed to various factors, including the illegal netting of brood and jatka hilsa, the widespread use of prohibited gillnets, and the impact of climate change, particularly manifested in erratic rainfall patterns. While the total hilsa catch continues to increase consistently, the diminishing growth rate in hilsa production demands a closer examination. Therefore, it becomes imperative to delve into the study of hilsa's feeding habitat and feeding biology. A comprehensive understanding of the hilsa's predominant food preferences holds great promise for developing a nuanced conservation strategy, particularly focusing on specific plankton species. By identifying and prioritizing these preferred planktonic organisms, we can optimize conservation efforts to sustain their availability. This approach is crucial for ensuring a continuous and robust nutrient supply for hilsa fish, potentially addressing and mitigating the observed decline in production growth. The present study aimed to provide comprehensive knowledge about the feeding biology of *T. ilisha* by interlinking among eco-climatic factors, plankton abundance in the water column, plankton ingestion in the gut, and the gonadosomatic index.

The objectives of this current study are as follows: (i) to analyze the spatial variation of plankton abundance in both water and gut throughout the year, taking into account key ecological and climatic factors; (ii) to calculate selectivity indices to discern the occurrence of selective ingestion activities; and (iii) to deepen our understanding of the complex relationships among eco-climatic factors, the gonadosomatic index, plankton dynamics, and the selective feeding behavior of *T. ilisha* by integrating all available datasets. This study has met current demands and the results will be beneficial for academics, researchers, and other individuals involved in related pursuits.

## 2. Materials and Methods

### 2.1. Ethical Statement

This study considered the sampling of hilsa fish from different locations of the river along with other activities. Animal scientific procedures were carefully followed during these activities with prior approval from the Animal Welfare and Ethics Committee of the Bangladesh Agricultural University (BAU) (Ref. No. BAURES/ESRC/FISH-11/2022).

### 2.2. Study Area

The study was conducted in six habitats across Bangladesh, as indicated in Table 1. These habitats were categorized according to geographical areas, variations in salinity, and abundance of hilsa. The six hilsa habitats also include different characteristics. The Gaglajur Haor (wetland) and Kali River are upstream habitats, while the Meghna River, Padma River, Tetulia River, and Bay of Bengal are downstream habitats. The Meghna estuary

contains a mixture of fresh and salt water, while the Bay of Bengal is the only habitat with a significant saline water presence. Figure 1 denoted the study area of the country map.

**Table 1.** Longitude and latitude of sampling sites.

| Location | GPS Coordinates | | |
| --- | --- | --- | --- |
| | Site 1 | Site 2 | Site 3 |
| Gaglajur Haor, Netrokona | 24.803643, 91.107437 | 24.804359, 91.107201 | 24.802772, 91.107636 |
| Tetulia River, Barisal | 22.132731, 90.490980 | 22.148945, 90.501136 | 22.160998, 90.503075 |
| Kali River, Kishoregonj | 24.129192, 90.931914 | 24.130202, 90.931757 | 24.130536, 90.931705 |
| Meghna River, Chandpur | 24.154072, 90.935592 | 23.314200, 90.614044 | 23.285916, 90.636677 |
| Padma River | 23.926921, 89.257314 | 23.920214, 89.309277 | 23.942873, 89.321573 |
| Bay of Bengal, Cox's Bazar | 21.520980, 91.829054 | 21.496385, 91.855833 | 21.428648, 91.948873 |

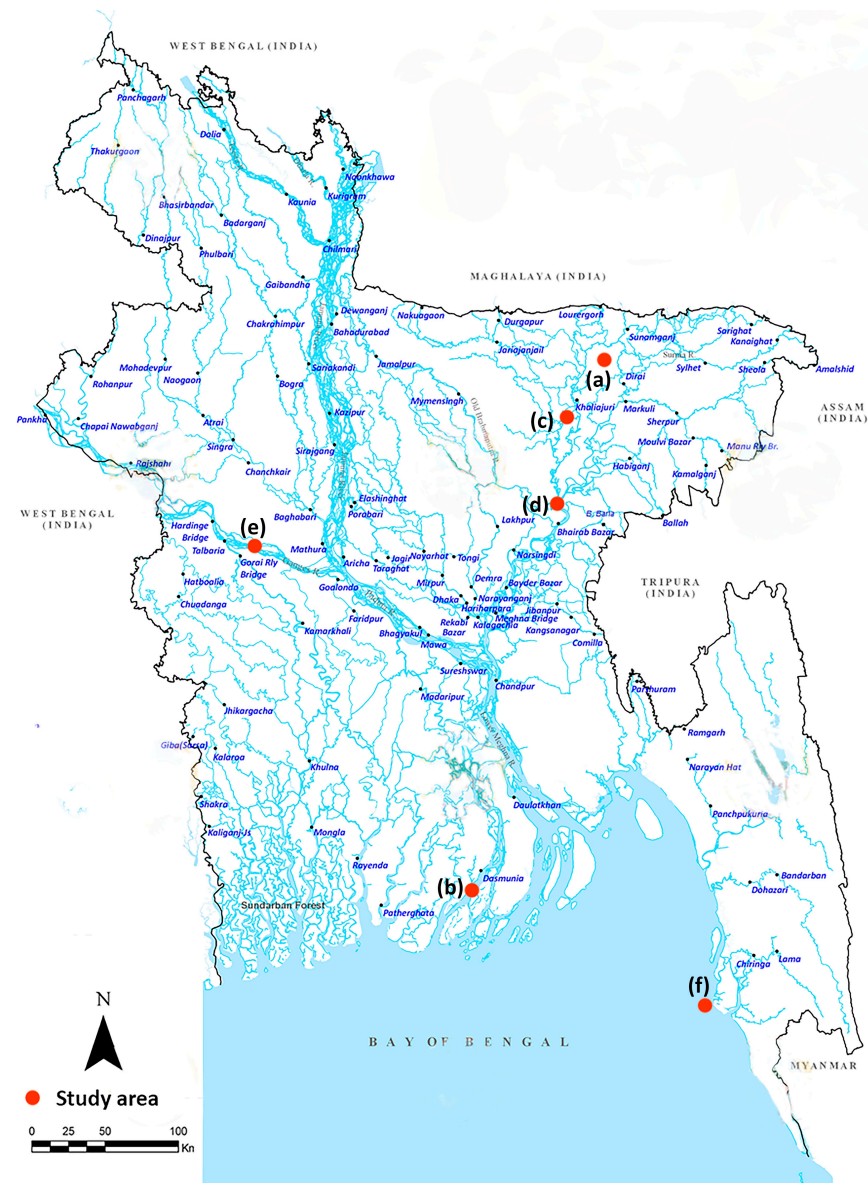

**Figure 1.** Bangladesh map showing the study area. (**a**) Gaglajur Haor, Netrokona, (**b**) Tetulia River, Barisal, (**c**) Kali River, Kishoregonj, (**d**) Meghna River, Chandpur, (**e**) Padma River, and (**f**) Bay of Bengal, Cox's Bazar.

*2.3. Data Collection*

A comprehensive mixed-methods approach was employed for data collection, drawing from both published and unpublished sources in the same location and context during the same study period [22–26]. This approach, which incorporates qualitative and quantitative research methods, aims to comprehensively explore a research hypothesis. The present study was part of a collaborative project titled "Studies on the causes of gonadal development in small-size Hilsa: Assessment of the factors associated with the early gonadal development of Hilsa" (Grant number: 2017/1201/BFRI), which was funded by the Bangladesh Fisheries Research Institute (BFRI). Overall, studies were conducted in six different areas of Bangladesh from January to December 2018. Under this project, various data, including climatic factors, water quality parameters, fish age, fish growth, Gonadosomatic Index (GSI), and plankton abundance in water and hilsa gut (both qualitative and quantitative) were collected using appropriate methods. The present study utilized data on water quality parameters and GSI from previously published articles which were part of the same research project [6,27]. These studies were significant in highlighting the findings of the current research. The present study utilized climatic data and plankton abundance in both water and hilsa gut (number/L) (Sections 2.3.1–2.3.3) to meet the growing demand of multifactorial linkage eco-climatic factors, plankton abundance, and gonadal maturation of Hilsa populations in Bangladesh.

2.3.1. Determination of Plankton Abundance in the Water of Six Diverse Habitats of *T. ilisha*

For the quantitative and qualitative analysis of plankton abundance in various habitats, plankton samples were collected in triplicate at each sampling site. Sampling occurred twice daily (at 12:00 p.m. and 06:00 p.m.) at each location. A plankton net, with a mesh size of 10 μm for phytoplankton and 90 μm for zooplankton, was employed to collect samples from the sub-surface layer of the water column. Following collection, the samples were preserved in 5% neutral buffered formalin within plastic bottles and transported to the Fisheries Biology and Genetics Laboratory of Bangladesh Agricultural University for subsequent analysis. A total of thirty samples were collected from each habitat for analysis. In the laboratory, 1 mL of the concentrated plankton sample was extracted and placed on a Sedgwick-Rafter (S-R) counting cell (Model 550, Fisons, Ipswich, UK). The S-R cell is a specialized slide featuring a counting chamber measuring 55 mm in length, 20 mm in width, and 1 mm in depth, with a total volume of 1 mL. This chamber is divided into 1000 fields, each with a volume of 0.001 mL. Plankton samples were examined under a light microscope for identification and counting. The abundance of plankton was estimated by tallying the presence of plankton per focus of the microscopic field, following the appropriate guidelines [28]. To identify plankton up to the genus level, established keys from several previous studies were used [29–31]. Additionally, planktons were classified into different groups based on the classification system proposed by a previous study [32]. This comprehensive methodology ensures a thorough and precise assessment of plankton abundance and diversity in the studied habitats. The number of plankton in the S-R cell was derived from the following formula:

$$\text{Number/Liter} = \frac{C \times 1000 \text{ mm}^3}{L \times D \times W \times S}$$

where C = number of organisms counted, L = length of each stripe (S-R cell length) in mm, D = depth of each stripe in mm, W = width of each stripe in mm, and S = number of stripes.

2.3.2. Determination of Gut Plankton Abundance and Selective Feeding Indices of *T. ilisha*

To investigate plankton in the digestive tracts, 30 specimens were collected from each sampling site encompassing varying sizes (small, medium, and large) while maintaining a balanced sex ratio. The alimentary canals of the fish were dissected from the esophagus to the anus and then preserved in a solution of 10 percent buffered formalin. Following preservation, the contents of the digestive tract, obtained from the pyloric stomach to the

gizzard of each fish, were subsequently dissolved in water. The presence of food organisms, with a specific focus on plankton, was scrutinized using a compound microscope (Olympus BH2, Tokyo, Japan) equipped with a camera at magnifications of $10 \times 40$ and $16 \times 4$. Sand, debris, and digested or unidentified plankton were not considered in counting during gut content analysis. The microscope, featuring both bright field and phase contrast illumination, facilitated a thorough examination of the samples. For both qualitative and quantitative analysis, a Sedgwick-Rafter counting cell was utilized. Plankton specimens were identified up to the genus level through the application of identification keys from several previous studies [29–31]. Furthermore, classification into distinct groups was carried out following the system reported by a previous study [32]. This methodology ensured a detailed exploration of plankton composition in the gut contents of the examined fish. The number of plankton in the S-R cell was derived from the following formula:

$$\text{Number/Liter} == \frac{C \times 1000 \text{ mm}^3}{L \times D \times W \times S}$$

where C = number of organisms counted, L = length of each stripe (S-R cell length) in mm, D = depth of each stripe in mm, W = width of each stripe in mm, and S = number of stripes.

The recorded food organisms were compared with the food organisms collected from the natural environment. Food preferences were analyzed for each location using Ivlev's food preference index, known as the 'electivity index' (E) [33]. The E value is determined using the following equation:

$$E = \frac{P_g - P_w}{P_g + P_w}$$

where Pg = percentage of a particular food organism in the gut and Pw = percentage of a particular food organism in the water. E values vary between –1 to +1. Positive values indicate selection for a certain food item while negative values indicate avoidance.

### 2.3.3. Collection of Climatic Variables, Water Quality Parameters, and GSI Data

The present study is a chronological continuation of two previous studies [6,21]. For determining the relationship among water quality, climatic factors, plankton abundance, gut content, and GSI, the water quality and GSI data were adopted from a study which has been published previously [6]. The data of climatic variables such as air temperature, rainfall, wind speed, and relative humidity for this study were collected from the National Oceanic and Atmospheric Administration (NOAA) Climate Data Online (CDO) database. This database is one of the largest sources of historical weather and climate data and is maintained by the National Centers for Environmental Information (NCEI). Data was collected through the CDO web interface by selecting the geographical location in decimal degrees of the Meghna River, the Padma River, the Tetulia River, the Bay of Bengal, the Gaglajur Haor, and the Kali River with the required study period (Table 1). The data was then downloaded in CSV (Comma Separated Values) format. The data was imported into a spreadsheet program and was thoroughly checked for missing values and outliers.

### 2.4. Statistical Analysis

All statistical analyses were conducted using R, version 3.6.1 [34]. The assumptions of normal distributions of all datasets were checked with the Shapiro–Wilk test, and homogeneity of variances were checked with Levene's test using the 'one way tests' package [35]. The univariate analysis of variance (ANOVA) was applied using the 'car' package [36]. The correlation matrix was employed using the 'corrplot' package. The cor() function was used to calculate Pearson's correlation coefficient using the 'number' and 'color' methods. The linear regression model was performed by using 'stats' package. The heat map with cluster analysis was produced using the 'dendextend' and 'pheatmap' packages [37]. The principal component analysis (PCA) for all datasets was performed using the 'FactoMineR' and 'Factoextra' packages [38,39]. All the plots were made using the 'ggplot2' package [40].

## 3. Results

### 3.1. Assessment of Climatic and Water Quality Parameters in the Six Diverse Habitats of T. ilisha

The annual highest average air temperature of 26.24 ± 3.01 °C was recorded in the Bay of Bengal, while the lowest temperature of 24.87 ± 4.00 °C was in Gaglajur Haor. The Tetulia River had the highest relative humidity at 80.34% (±9.21), while the Padma River had the lowest at 75.89% (±14.82). Rainfall, an important climatic factor influencing gonadal maturation, was highest in the Bay of Bengal (26.00 ± 30.88 mm), while the lowest was in the Padma River (7.43 ± 6.14 mm). The Tetulia River exhibited the highest wind speed at 8.13 ± 2.31 km/h, while the Meghna River had the lowest wind speed at 3.08 ± 0.87 km/h (Table 2).

**Table 2.** Analysis of the climatic variables and water quality parameters in the six diverse habitats of *T. ilisha*.

| Parameters | Bay of Bengal | Gaglajur Haor | Kali River | Meghna River | Padma River | Tetulia River | F Value | *p* Value | Level of Sig. |
|---|---|---|---|---|---|---|---|---|---|
| Air temperature (°C) | 26.24 ± 3.01 | 24.87 ± 4.00 | 25.60 ± 4.16 | 25.62 ± 3.95 | 25.78 ± 4.34 | 25.65 ± 3.85 | 0.152 | 0.979 | NS |
| Relative humidity (%) | 79.31 ± 6.89 | 77.69 ± 12.19 | 76.43 ± 14.18 | 78.87 ± 11.44 | 75.89 ± 14.82 | 80.34 ± 9.21 | 0.258 | 0.934 | NS |
| Rainfall (mm) | 26.00 ± 30.88 | 15.29 ± 16.44 | 8.18 ± 7.55 | 9.98 ± 9.72 | 7.43 ± 6.14 | 14.24 ± 14.74 | 2.09 | 0.076 | NS |
| Wind speed (km/h) | 7.97 ± 2.16 | 6.24 ± 1.66 | 7.17 ± 1.78 | 3.08 ± 0.87 | 7.41 ± 1.79 | 8.13 ± 2.31 | 12.84 | 0.000 | ** |
| Water temperature (°C) | 27.46 ± 0.84 | 27.47 ± 0.83 | 28.15 ± 1.32 | 27.24 ± 0.58 | 27.60 ± 0.74 | 27.61 ± 1.11 | 1.29 | 0.278 | NS |
| DO (ppm) | 6.43 ± 0.10 | 8.43 ± 0.15 | 7.56 ± 0.12 | 11.57 ± 0.43 | 12.58 ± 0.30 | 10.56 ± 0.14 | 1211.94 | 0.000 | ** |
| pH | 6.62 ± 0.09 | 9.34 ± 0.17 | 9.41 ± 0.12 | 8.21 ± 0.20 | 8.43 ± 0.10 | 7.59 ± 0.18 | 615.07 | 0.000 | ** |
| Salinity (ppm) | 29.67 ± 0.48 | 0.00 ± 0.00 | 0.00 ± 0.00 | 0.00 ± 0.00 | 0.00 ± 0.00 | 0.43 ± 0.17 | 40,536.02 | 0.000 | ** |

Values presented as mean ± standard deviation. NS = Not significant, ** = significant in 1% significance level.

### 3.2. Determination of Plankton Abundance in the Six Diverse Habitats of T. ilisha

During the study period, a numerical analysis revealed a total of 13 plankton groups, including 6 phytoplankton and 7 zooplankton groups, comprising a total of 101 planktonic genera/species. Of these, 74 belonged to phytoplankton and 27 to zooplankton. The groups identified were Bacillariophyceae (24), Chlorophyceae (26), Cyanophyceae (15), Euglenophyceae (2), Xanthophyceae (2), Dinophyceae (4), Copepoda (8), Cladocera (5), Rotifera (7), Protozoa (2), Polychaeta (1), Gastropoda (1), and Hydrozoa (3). These findings were recorded in the water column from the Bay of Bengal, the Meghna River, the Padma River, the Tetulia River, the Gaglajur Haor, and the Kali River (Figure 2 and Table 3). In every location, the dominant phytoplankton groups in the water column were Bacillariophyceae, Chlorophyceae, Cyanophyceae, and Euglenophyceae.

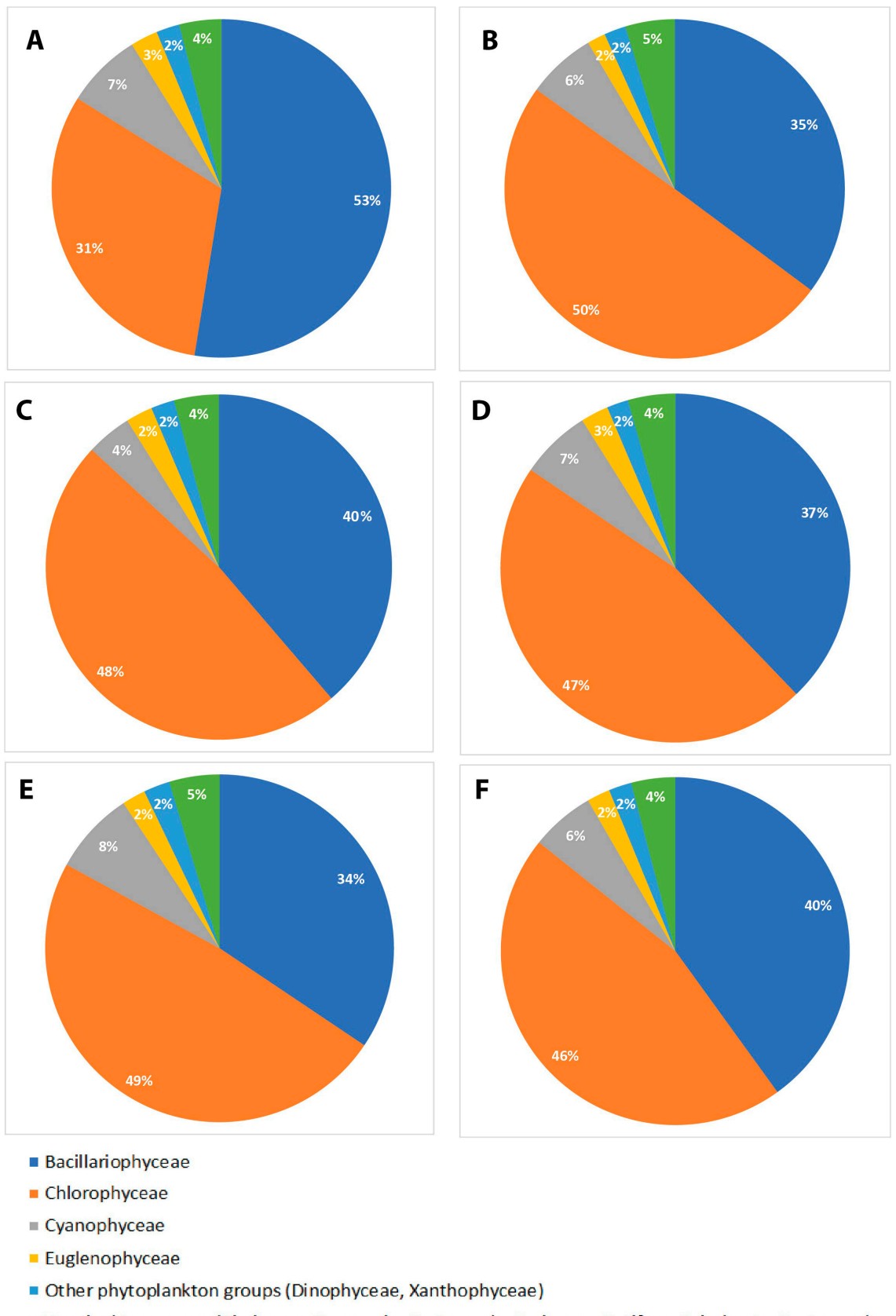

**Figure 2.** Percentage composition of plankton abundance (number/L) in water. (**A**) Bay of Bengal; (**B**) Gaglajur Haor; (**C**) Kali River; (**D**) Meghna River; (**E**) Padma River; (**F**) Tetulia River.

**Table 3.** Plankton genera observed in the six different habitats of *T. ilisha*.

| Class | |
|---|---|
| **Phytoplankton** | **Genus/Species** |
| Bacillariophyceae (24 spp.) | *Amphora* sp., *Anomoeoneis* sp., *Asterionella* sp., *Bacillaria* sp., *Coscinodiscus* sp., *Cyclotella* sp., *Cymbella* sp., *Diatoma* sp., *Diatomella* sp., *Ditylum* sp., *Eunotia* sp., *Eucampia* sp., *Fragillaria* sp., *Gomphonema* sp., *Gyrosigma* sp., *Navicula* sp., *Nitzschia* sp., *Pleurosigma* sp., *Rhizosolenia* sp., *Surirella* sp., *Synedra* sp., *Skeletonema* sp., *Tabellaria* sp., *Thalassionema* sp. |
| Chlorophyceae (26 spp.) | *Actinastrus* sp., *Ankistrodesmus* sp., *Botryococcus* sp., *Chlorella* sp., *Closterium* sp., *Coelastrum* sp., *Micractinium* sp., *Microspora* sp., *Muogeotia* sp., *Oedogonium* sp., *Oocystis* sp., *Palmella* sp., *Pediastrum* sp., *Pleorococcus* sp., *Pteromonas* sp., *Scenedesmus* sp., *Selenestrum* sp., *Spirogyra* sp., *Staurastrum* sp., *Stichococcus* sp., *Tetraedron* sp., *Triceratium* sp., *Ulothrix* sp., *Uroglena* sp., *Volvox* sp., *Zygnema* sp. |
| Cyanophyceae (15 spp.) | *Anabaena* sp., *Aphanizomenon* sp., *Aphanocapsa* sp., *Chroococcus* sp., *Cosmarium* sp., *Closterium* sp., *Dictyophimus* sp., *Desmidium* sp., *Gomphosphaeria* sp., *Merismopedium* sp., *Microcystis* sp., *Microspora* sp., *Nostoc* sp., *Oscillatoria* sp., *Spirulina* sp. |
| Euglenophyceae (2 spp.) | *Euglena* sp., *Phacus* sp. |
| Xanthophyceae (2 spp.) | *Botrydium* sp., *Tribonema* sp. |
| Dinophyceae (4 spp.) | *Ceratium* sp., *Gyrodinium* sp., *Dinophysis* sp., *Peridinium* sp |
| Zooplankton | |
| Copepoda (8 spp.) | *Acartia* sp., *Calanus* sp., *Calanopia* sp., *Cyclops* sp., *Diaptomus* sp., *Laptodora* sp., *Naupleus* sp., *Paracartia* sp. |
| Cladocera (5 spp.) | *Bosmina* sp., *Diaphanosoma* sp., *Daphnia* sp., *Moina* sp., *Sida* sp. |
| Rotifera (7 spp.) | *Asplanchna* sp., *Brachionus* sp., *Filinia* sp., *Hexarthra* sp., *Keratilla* sp., *Poliarthra* sp., *Trichocerca* sp. |
| Protozoa (2 spp.) | *Difflugia* sp., *Favella* sp. |
| Polychaeta (1 spp.) | *Pedinosoma* sp. |
| Gastropoda (1 spp.) | *Notobranchaea* sp. |
| Hydrozoa (3 spp.) | *Aglaura* sp., *Chelophyes* sp., *Liriope* sp. |

*3.3. Determination of Gut Plankton Abundance and Selective Feeding Indices of T. ilisha*

The quantitative and qualitative study of hilsa gut contents from various locations revealed that Bacillariophyceae (diatoms), Chlorophyceae (green algae), and crustaceans (Copepoda and Cladocera) constituted the major portion of their diet (Figure 3). Considerable amounts of Chlorophyceae and Bacillariophyceae, 35% and 57%, respectively, were observed in the Bay of Bengal (Figure 3A); 51% and 38%, respectively, in the Gaglajur Haor (Figure 3B); 50% and 40%, respectively, in the Kali River (Figure 3C); 53% and 39%, respectively, in the Meghna River (Figure 3D); 49% and 41%, respectively, in the Padma River (Figure 3E); and 49% and 42%, respectively, in the Tetulia River (Figure 3F). These were found in the gut contents of hilsa.

Among the 101 genera of plankton present in the water column, Indian shad ingested 49 genera, including 14 of Chlorophyceae, 7 of Cyanophyceae, 12 of Bacillariophyceae, 1 of Euglenophyceae, 1 of Xanthophyceae, 2 of Dinophyceae, 2 of Copepoda, 2 of Cladocera, 3 of Rotifera, 1 of Protozoa, 1 of Polychaeta, 1 of Gastropoda, and 1 of Hydrozoa. The electivity index was used to determine the food preferences of hilsa. The overall electivity index results showed that hilsa preferred phytoplankton over zooplankton. Among the phytoplankton, Bacillariophyceae or diatoms (+0.04, +0.04, +0.02, +0.02, +0.09, and +0.02) and Chlorophyceae or green algae (+0.05, +0.01, +0.02, +0.06, +0.01, and +0.03) were preferred in the Bay of Bengal, the Gaglajur Haor, the Kali River, the Meghna River, the Padma River, and the Tetulia River, respectively (Table 4). In addition, these electivity indices indicated that Indian shad had a positive preference for Rotifera (+0.25) in the Bay of Bengal; Euglenophyceae (+0.01) and Dinophyceae (+0.20) in the Gaglajur Haor; Cyanophyceae (+0.09 and +0.03) in the Kali River and the Tetulia River, respectively; and

Dinophyceae (+0.02) in the Padma River. However, it was revealed that the majority of zooplankton groups were not preferred by *T. ilisha* (Table 4).

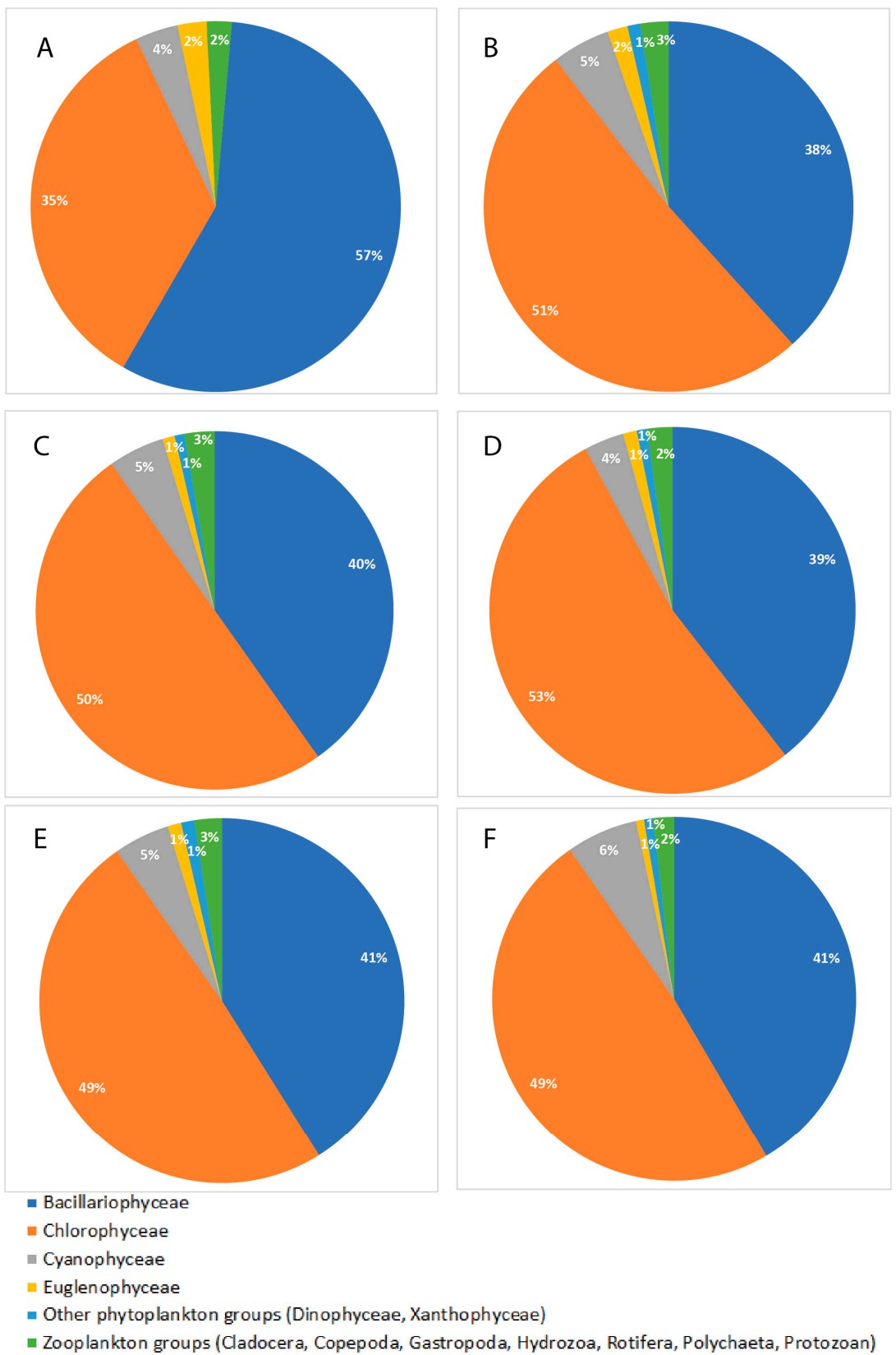

**Figure 3.** Percentage composition of plankton abundance (number/L) in the gut. (**A**) Bay of Bengal; (**B**) Gaglajur Haor; (**C**) Kali River; (**D**) Meghna River; (**E**) Padma River; (**F**) Tetulia River.

**Table 4.** Ivlev's electivity index values showing hilsa food preferences for different plankton groups in different hilsa habitats.

| Groups | Bay of Bengal | Gaglajur Haor | Kali River | Meghna River | Padma River | Tetulia River |
|---|---|---|---|---|---|---|
| Bacillariophyceae | 0.04 | 0.04 | 0.02 | 0.02 | 0.09 | 0.02 |
| Chlorophyceae | 0.05 | 0.01 | 0.02 | 0.06 | 0.01 | 0.03 |
| Cyanophyceae | −0.32 | −0.13 | 0.09 | −0.30 | −0.22 | 0.03 |
| Euglenophyceae | −0.06 | 0.01 | −0.39 | −0.36 | −0.29 | −0.50 |
| Dinophyceae | −0.59 | 0.20 | −0.24 | 0.24 | 0.02 | −0.27 |
| Xanthophyceae | −1.00 | −0.56 | −0.57 | −0.69 | −0.50 | −0.58 |
| Cladocera | −0.46 | −0.28 | −0.01 | −0.32 | −0.18 | −0.27 |
| Copepoda | −0.28 | −0.17 | −0.24 | −0.27 | −0.28 | −0.33 |
| Gastropoda | −0.24 | −0.37 | −0.41 | −0.46 | −0.67 | −0.69 |
| Hydrozoa | −0.80 | −0.60 | −0.59 | −0.76 | −0.63 | −0.65 |
| Rotifera | 0.25 | −0.34 | 0.05 | −0.11 | −0.04 | −0.52 |
| Polychaeta | −0.33 | −0.45 | −0.23 | −0.41 | −0.23 | −0.07 |
| Protozoan | −0.48 | −0.64 | −0.32 | −0.38 | −0.60 | −0.44 |

*3.4. Multivariate Analysis among the Eco-Climatic Variables, Plankton Abundance in Water, Gut, and GSI of T. ilisha*

According to the correlation matrix (Figure 4), significant correlations were found among the climatic variables, water quality parameters, plankton abundance in water and gut (annual average of both phytoplankton and zooplankton), and the gonadosomatic index of *T. ilisha*. A strong positive correlation (0.74) has been found between plankton abundance in the gut and GSI. Air temperature and water temperature are positively correlated (0.60). Relative humidity showed a strong positive correlation with air temperature (0.51), while wind speed revealed a positive correlation with both air temperature (0.49) and water temperature (0.49). Rainfall has a moderate positive correlation with both air temperature (0.49) and water temperature (0.43). On the other hand, a positive correlation (0.52) has been found between rainfall and relative humidity. There is a moderate negative relationship (−0.35) between plankton abundance in the gut and water temperature, and a positive relationship (0.74) with GSI. Plankton abundance in the water showed a positive relationship (0.07) with dissolved oxygen (DO). Salinity has a negative relationship (−0.62) with dissolved oxygen (DO) and (−0.76) with pH (Figure 4).

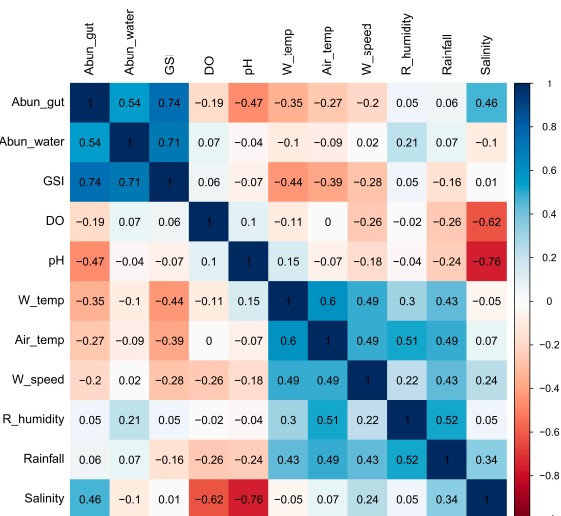

**Figure 4.** Correlations among climatic variables, water quality parameters, GSI, and plankton abundance in both gut and water based on Pearson's correlations. Color key: red and blue colors represent positive and negative correlations, respectively. Here, the variables' full names are Abun_gut—Plankton abundance in gut (Number/L); W_temp—Water temperature; Air_temp—Air temperature (°C); R_humidity—Relative humidity (%); W_speed—Wind speed (km/h); Abun_water—Plankton abundance in water (Number/L); Rainfall (mm); DO (ppm); and Salinity (ppm).

A heat map can directly display intragroup and intergroup similarities and differences. Cluster analysis is a tool used to evaluate differences through an algorithm that produces a dendrogram [41]. In this study, a heatmap cluster analysis with complete linkage was used to evaluate the similarities and differences of various plankton groups in six locations (the Meghna River, the Padma River, the Tetulia River, the Bay of Bengal, the Gaglajur Haor, and the Kali River) in relation to their sources (gut and water) and types (phytoplankton and zooplankton). The analysis involved column scaling with normalization. The results showed that the 13 plankton groups were classified into 2 types and 2 sources, observed in 6 locations, forming 4 clusters, and exhibiting significant spatial aggregation (Figure 5). Hierarchical clustering was employed using the complete linkage method with Euclidean distance as the similarity metric to determine the optimal number of clusters. In addition, the dendrogram for rows revealed that plankton abundance in water and gut clustered separately for six locations. The rows representing water sources showed more deep red color boxes than the columns representing gut sources, indicating a higher plankton abundance in the water sources than in the gut (Figure 5). In terms of plankton types, two distinct clusters were shown in the column dendrogram. The Chlorophyceae, Bacillariophyceae, Euglenophyceae, and Rotifera clustered together, while the rest of the plankton groups were in the other cluster (Figure 5). In the gut source cluster, small numbers of phytoplankton groups (cyanophyceae, Dinophyceae, and Xanthophyceae) and large numbers of zooplankton groups (Protozoan, Hydrozoa, Polychaeta, Cladocera, Gastropod, and Copepoda) were less dominant. Conversely, these groups showed high dominance in the water source cluster (Figure 5).

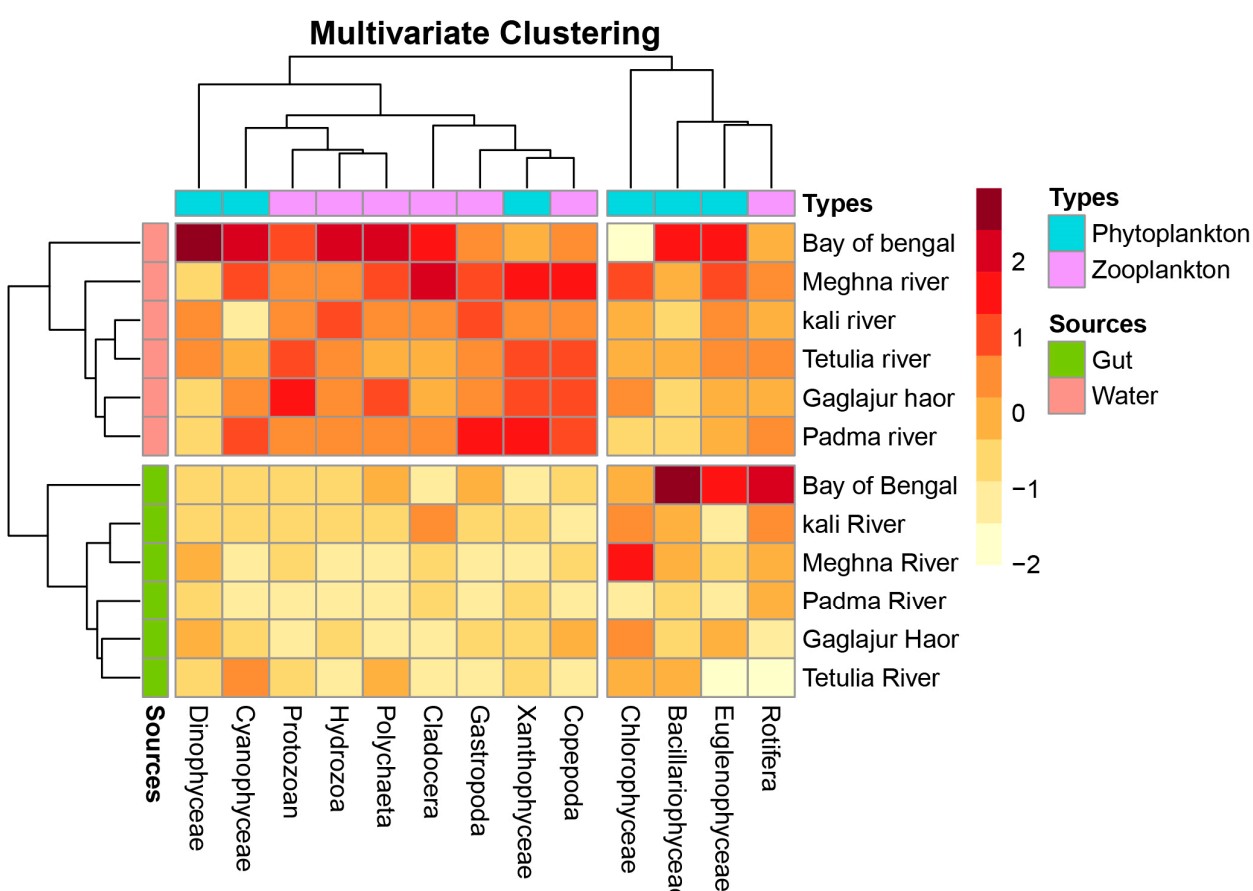

**Figure 5.** Heat map using Euclidean distance as the similarity measure and clustering plankton abundance both in water and gut from six different hilsa habitats based on the 'complete' method.

According to the principal component analysis (PCA), a significant relationship existed among the climatic variables, water quality parameters, plankton abundance in water and the gut, and gonadosomatic index of hilsa in six different habitats. The elbow of the scree plot is located in the fourth component and collectively explains 79.3% of the variance (Figure 6A). The initial component accounted for 29.52% of the variance (Figure 6A and Table 5). This component revealed the relationship between air temperature, water temperature, wind speed, rainfall, and GSI, either collectively or individually, with significant positive factor loadings (Figure 6B,C). The second component explained 25.11% of the variance (Figure 6A and Table 5) and was largely influenced by plankton abundance in the gut, pH, and salinity, either collectively or individually, as indicated by significant factor loadings (Figure 6B,D). From the PCA biplot, it is evident that plankton abundance in the gut had the highest contribution followed by GSI, water temperature, and air temperature (Figure 7). Monthly variations (Figure 8A) and station-based variations (Figure 8B) were revealed in the individual PCA plot. The samples from the Bay of Bengal clustered distinctly, showing higher salinity levels compared to the other five stations.

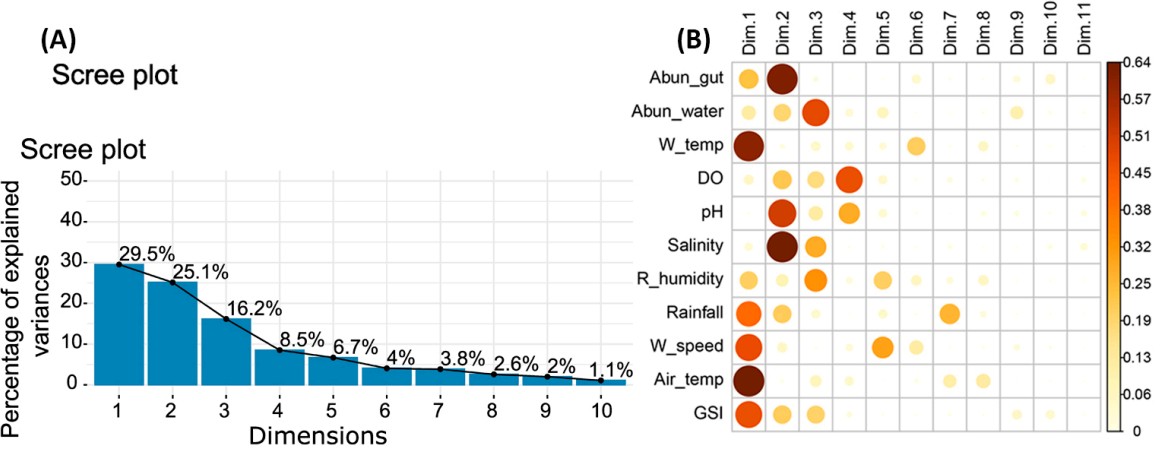

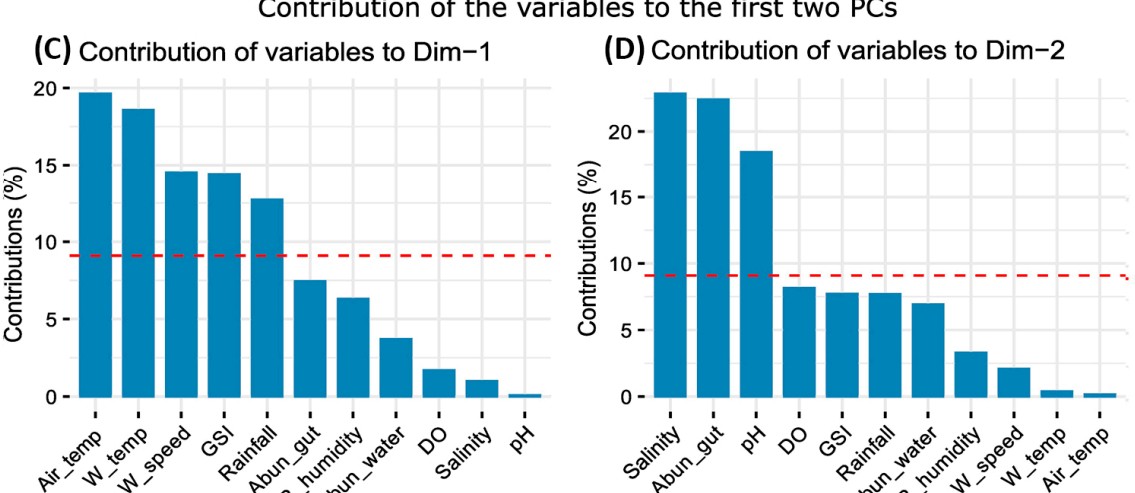

**Figure 6.** (**A**) Scree plot; (**B**) Component matrix; (**C**) Contribution of variables of first component; (**D**) Contribution of variables of second component. Here, the variables' full names are: Abun_gut: Plankton abundance in gut (Number/L); W_temp: Water temperature (°C); Air_temp: Air temperature (°C); R_humidity: Relative humidity (%); W_speed: Wind speed (km/h); Abun_water: Plankton abundance in water (Number/L); Rainfall (mm); DO (ppm); Salinity (ppm). The crossing of the parameters over the dotline is considered a significant contribution.

**Table 5.** Rotated component matrix for eco-climatic factors and plankton abundance, including eigenvalue and variance.

| Variable | PC1 | PC2 | PC3 |
|---|---|---|---|
| Plankton abundance in gut (Number/L) | −0.27 | 0.57 | −0.08 |
| Plankton abundance in water (Number/L) | −0.19 | 0.26 | −0.52 |
| Water temperature (°C) | 0.59 | −0.06 | −0.17 |
| DO (ppm) | −0.13 | −0.29 | −0.31 |
| pH | −0.02 | −0.43 | −0.26 |
| Salinity (ppm) | 0.09 | 0.58 | 0.4 |
| Relative humidity (%) | 0.25 | 0.18 | −0.43 |
| Rainfall (mm) | 0.36 | 0.27 | −0.15 |
| Wind speed (km/h) | 0.38 | 0.14 | −0.02 |
| Air temperature (°C) | 0.54 | 0.03 | −0.21 |
| GSI | −0.38 | 0.27 | −0.34 |
| Eigenvalue | 3.24 | 2.76 | 1.78 |
| Variance (%) | 29.52 | 25.11 | 16.16 |
| Cumulative variance (%) | 29.52 | 54.64 | 70.79 |

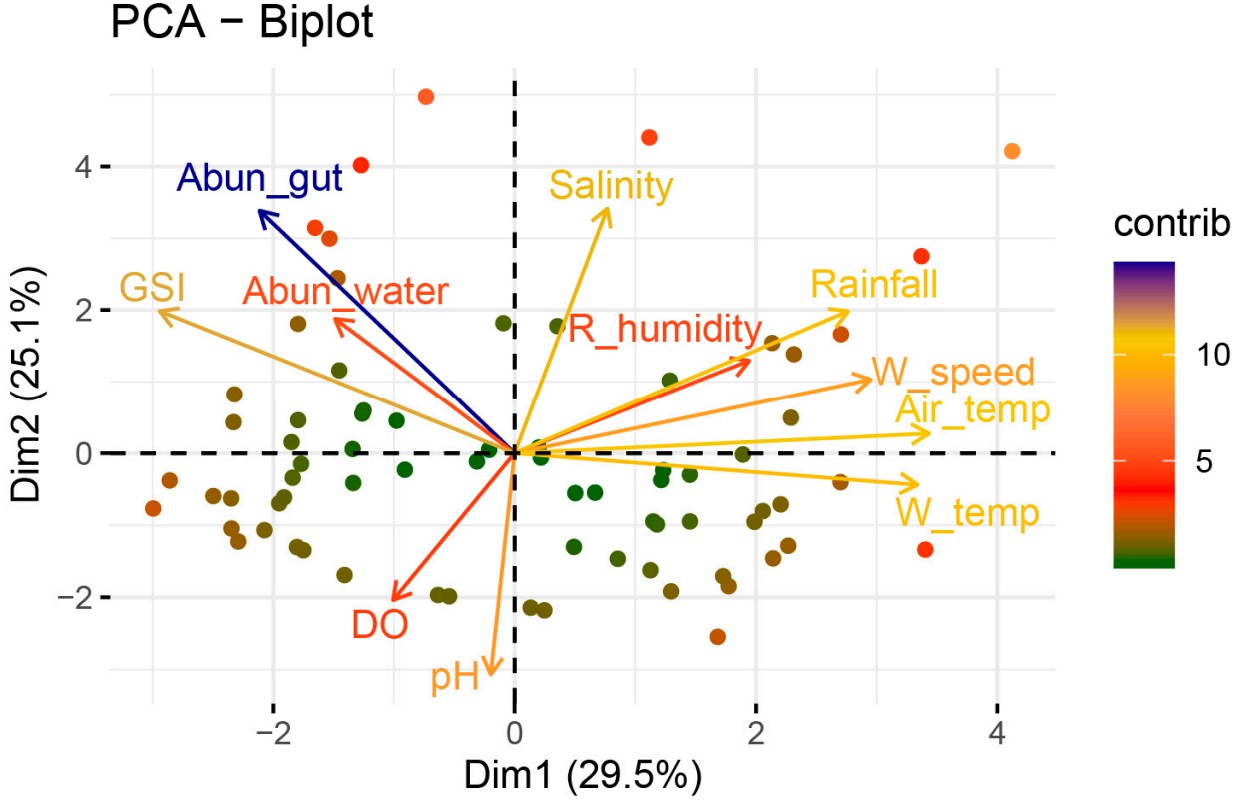

**Figure 7.** PCA biplot of multivariate relationship among plankton abundance and GSI in relation with climatic and water quality parameters. Here, the variables' full names are Abun_gut—Plankton abundance in gut; W_temp—Water temperature; Air_temp—Air temperature; R_humidity—Relative humidity; W_speed—Wind speed; and Abun_water—Plankton abundance in water.

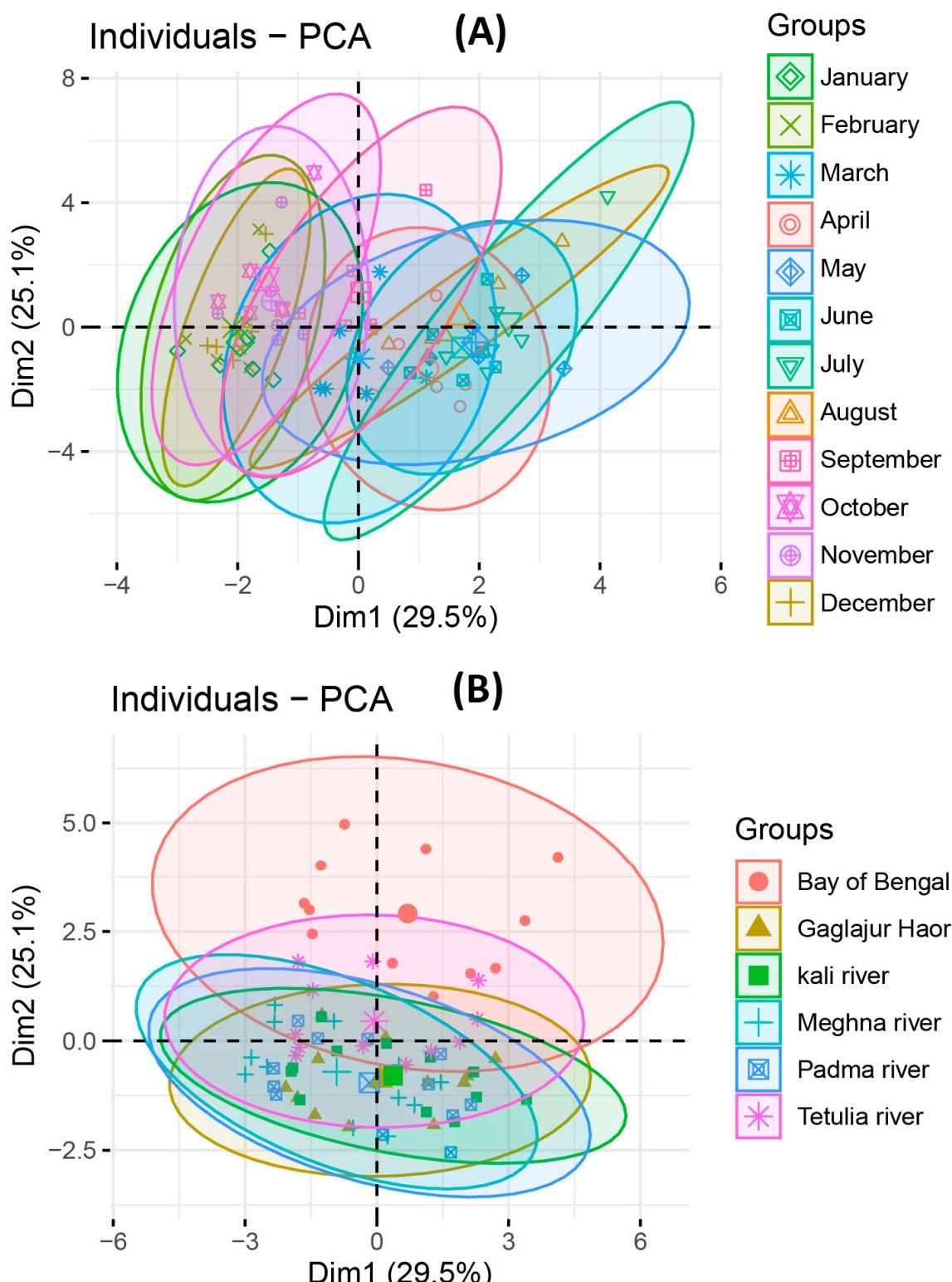

**Figure 8.** Individual PCA clusters of variables based on (**A**) months (**B**) stations.

*3.5. Regression Analysis between Plankton Abundance in Gut and GSI of T. ilisha*

The linear regression model has been used to assess the impact of gut plankton abundance on gonadal maturation, with plankton abundance in the gut as the independent variable and GSI as the dependent variable. The upward trend of the regression line, with R > 0.75 in all habitats, indicates a significant positive impact of plankton abundance in the gut on the GSI of *T. ilisha* (Figure 9). In all habitats, the *p* values were lower than 0.05,

indicating a significant dependency of gonadal maturation of *T. ilisha* on the abundance of plankton in the gut (number/L).

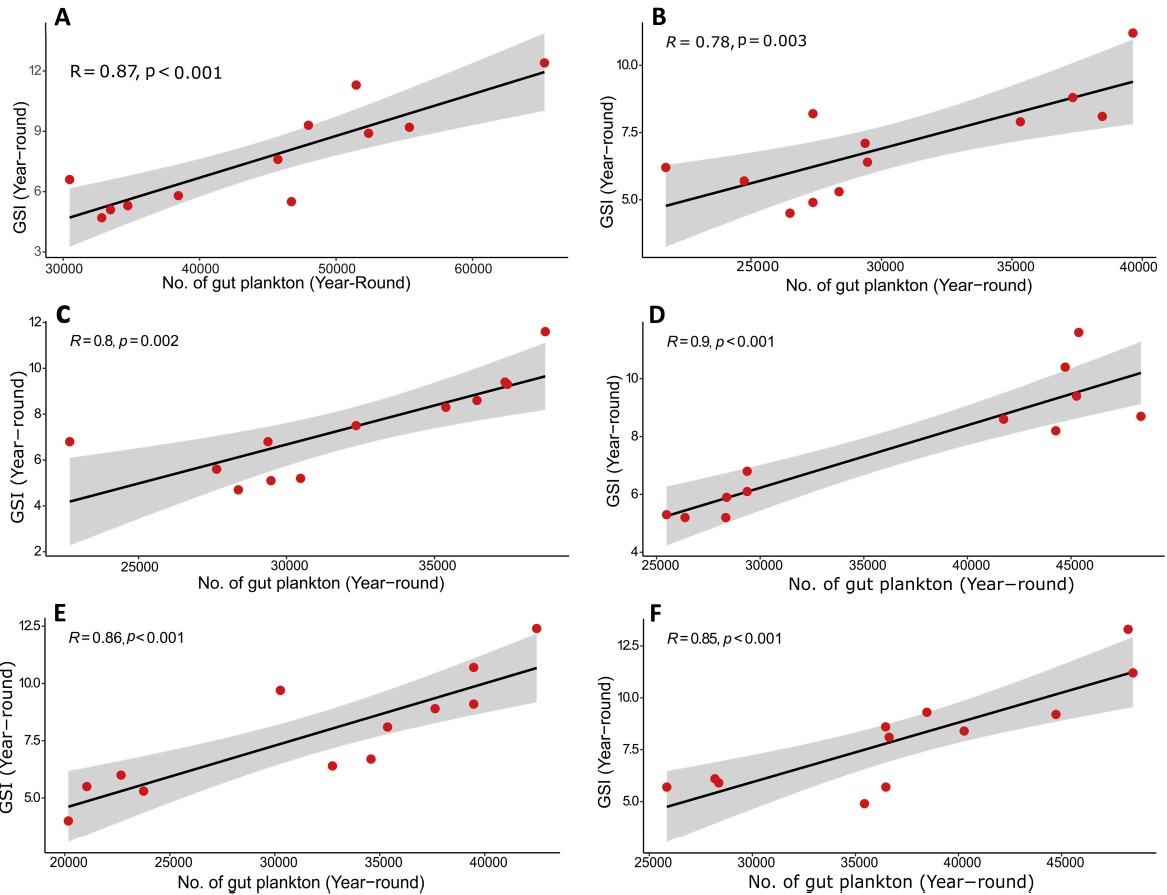

**Figure 9.** Scatter diagrams with linear regression analyses. (**A**) Bay of Bengal; (**B**) Gaglajur Haor; (**C**) Kali River; (**D**) Meghna River; (**E**) Padma River; (**F**) Tetulia River. The black line in the graph signifies the regression line, with the grey color denoting the 95% confidence interval, and individual data points are visually represented as red dots.

## 4. Discussion

The study examined different climatic variables, including air temperature (°C), relative humidity (%), rainfall (mm), and wind speed (km/h), in six distinct hilsa habitats in Bangladesh. The Bay of Bengal had the highest average air temperature (26.24 ± 3.01 °C), rainfall (26.00 ± 30.88 mm), and salinity (29.67 ± 0.48 ppm), while the Gaglajur Haor had the lowest air temperature (24.87 ± 4.00 °C). The Tetulia River had the highest average wind speed and relative humidity, while the Padma River had the highest average DO levels. The Kali River had the highest average water temperature and pH. Monthly variations were also observed in these environmental factors. The study found significant differences in wind speed, DO, pH, and salinity among the six hilsa habitats. The variations in climatic conditions in different river habitats are influenced by altitude, proximity to the ocean, and local weather patterns. These factors result in significant variations in climatic variables. Understanding these variations is important for predicting the impacts of climate change [19,42–44]. The changes in climatic variables impact water quality parameters, which are consistent with numerous previous findings. Water temperature is significantly affected by air temperature, as warmer air temperatures result in higher water temperatures [45–47]. This is because water absorbs heat from the surrounding air [45,48,49]. Similarly, wind speed can also influence water temperature by causing mixing of the water column, which can lead to changes in temperature [50–52]. Dissolved oxygen (DO) levels in

water are affected by temperature and salinity [6,53–56]. Higher temperatures lead to lower dissolved oxygen levels because warm water holds less oxygen than cold water [6]. Salinity also affects dissolved oxygen (DO) levels, with lower salinity leading to higher DO levels because saltwater holds less oxygen than freshwater [6,57,58]. The pH is influenced by various factors, including atmospheric $CO_2$ concentrations, temperature, and the presence of other dissolved substances [59,60]. Higher atmospheric $CO_2$ concentrations can lead to lower pH levels in water, as can higher temperatures [59]. The presence of dissolved substances, such as acids or bases, can also affect the pH levels in water [61]. Salinity levels in water are mainly influenced by the influx of freshwater into the system, such as from rainfall or rivers [62]. Higher levels of rainfall can result in lower salinity levels, whereas lower levels of rainfall can lead to higher salinity levels [62,63]. High humidity indirectly impacts water quality by influencing the hydrological cycle, resulting in increased precipitation and the influx of water into water bodies [64,65].

The present study highlighted the dominant phytoplankton groups as Bacillariophyceae (34–53%), Chlorophyceae (31–50%), Cyanophyceae (4–8%), and Euglenophyceae (1–3%). The dominant zooplankton groups were Copepoda, Rotifera, and Cladocera. The composition of dominant groups varied among the six hilsa locations, with Bacillariophyceae and Chlorophyceae being the most prevalent phytoplankton groups, and Maxillopoda and Branchiopoda being common zooplankton groups in the Bay of Bengal. The Bay of Bengal has been found to have the highest plankton diversity, with climatic and water quality parameters that are suitable for this diversity. The recent findings are significantly consistent with a previous report in which the authors observed that plankton abundance appears to be stimulated during October to December by favorable conditions such as bright sunlight with less cloud cover and estuarine mechanisms [66]. Air temperature, relative humidity, rainfall, and wind speed can all have significant impacts on the plankton population in rivers [12]. Air temperature and relative humidity can directly and indirectly affect plankton, while rainfall can impact plankton through dilution, nutrient input, and physical disturbance [67–69]. Wind speed can affect the distribution and abundance of plankton through mechanisms such as mixing, transport, oxygenation, and temperature changes [70–72]. The overall impact on plankton will be influenced by the specific environmental conditions in the river ecosystem, including the preferred temperature range of different species and the existing nutrient availability [73,74]. Higher water temperatures promote faster growth and reproduction of plankton but can also lead to thermal stress and mortality in some plankton species [75,76]. Plankton require oxygen to survive, and low levels of dissolved oxygen can restrict their growth and distribution [77,78]. Changes in pH can affect nutrient availability and the solubility of contaminants that can harm plankton [71,74,79]. Salinity affects the availability of nutrients and water density. Some species of plankton are adapted to high salinity conditions, while others are adapted to freshwater conditions [70]. A study demonstrated that salinity significantly influenced the type of plankton present in lakes, leading to a higher abundance of salt-tolerant species and reduced species diversity, consistent with findings in other saline lakes [80–82]. The zooplankton species association was also similar to those found in other saline lakes in the region [83,84]. This was characterized by the dominance of rotifers and low copepod abundances, mainly consisting of cyclopoids. This pattern is commonly observed in eutrophic environments with high predation pressure from omnivorous planktivorous fish such as *Odonthesthes* sp. [85]. The abundance and biomass dynamics of plankton were influenced by various physicochemical drivers [70]. Seasonal variations in temperature were found to influence changes in abundance, with zooplankton being more abundant during warmer seasons and phytoplankton during colder months [70]. This was different from other shallow lakes in the Pampas region, where phytoplankton blooms are mainly observed in spring and summer. The composition and abundance of phytoplankton in the saline lakes were similar to those reported in other studies, as indicated by previous studies [80,86].

The present study also analyzed the gonadosomatic index (GSI) values of hilsa in six different habitats over the course of a year to assess gonadal development. The Padma River had the highest average GSI value (7.73 ± 2.50), followed by the Tetulia River (8.03 ± 2.53) and the Bay of Bengal (7.64 ± 2.57), while the Gaglajur Haor had the lowest GSI (7.03 ± 1.92). October was identified as the primary spawning season across all habitats, with the highest GSI values consistently observed during this month. The Tetulia River had the highest GSI values in August and October, indicating favorable spawning conditions during these months. Climatic factors such as air temperature, relative humidity, rainfall, and wind speed can indirectly affect hilsa gonad maturation in rivers by influencing water temperature, dissolved oxygen levels, pH, and salinity [87–89]. Relative humidity can also affect water temperature, as higher humidity can lead to cooler temperatures, while lower humidity can lead to warmer temperatures [63,67]. Heavy rainfall can impact the availability of hilsa food and lead to reduced levels of dissolved oxygen, while wind speed can affect water mixing and oxygenation [12,87,90]. The development of gonads in hilsa fish is strongly influenced by various water quality parameters, including water temperature, dissolved oxygen levels, pH, and salinity [21]. The optimal temperature for gonadal development is between 26 °C and 30 °C. If the temperature falls outside this range, it can delay or accelerate the development of gonads, affecting the timing of spawning. Previous studies have suggested that an increase in seawater temperature can accelerate the onset of gonadal maturity in hilsa fish [91,92]. With the Bay of Bengal experiencing a steady rise in temperature [93], it is probable that hilsa in this region may experience early gonadal maturity. A study reported that spawning begins at temperatures of 24–29 °C, with dissolved oxygen levels of 6.5 to 8.5 and pH ranging from 7.6 to 7.8 [94]. Similarly, insufficient levels of dissolved oxygen can induce stress and mortality in fish and decrease the rate of gonadal development [95]. Maintaining sufficient levels of dissolved oxygen is crucial for the healthy development of gonads in hilsa [95]. The pH of water can also affect the activity of enzymes involved in gonadal development, leading to abnormalities and delayed maturation of hilsa [96–98]. An appropriate salinity level during migration is essential for successful gonadal development in hilsa, as it is an anadromous fish species that spends most of its life in freshwater but migrates to estuaries and coastal areas to spawn [3,9,99].

As Hilsa prefer phytoplankton and zooplankton, these play a vital role in their growth and development. Phytoplankton, tiny aquatic plants that form the foundation of the aquatic food web, are essential for the survival and growth of zooplankton, tiny aquatic animals that feed on phytoplankton [9,100,101]. Hilsa, a type of fish found in the river waters of Bangladesh, rely on zooplankton for food, and their gonad maturation can be influenced by the availability and quality of zooplankton [100,101]. An abundance of phytoplankton leads to an increase in zooplankton populations, which in turn provides a food source for hilsa [52,102,103]. The availability and quality of phytoplankton and zooplankton can affect the growth rate, body condition, and reproductive success of hilsa [100,101,104]. The availability of phytoplankton and zooplankton in Bangladesh is influenced by various factors such as nutrient inputs from agriculture and industry, river flow, temperature, and climate change. These factors can cause changes in the abundance and composition of phytoplankton and zooplankton, leading to cascading effects on the entire food web, including hilsa [105–108]. Phytoplankton and zooplankton contain a variety of nutrients, including proteins for tissue growth and repair, carbohydrates for energy, lipids for insulation and energy storage, vitamins such as A, D, and B12 for various bodily functions, minerals like calcium, iron, and magnesium essential for bone health, blood circulation, and nerve function, and omega-3 fatty acids that support brain function and can reduce the risk of heart disease [9,103,109–112]. It is noteworthy that the nutrient composition of these organisms can fluctuate based on the species and environmental factors, leading to variations in their nutrient content in the rivers of Bangladesh [110,111]. The present study analyzed the gut contents of hilsa from six different habitats and identified 18 groups of plankton. The Bay of Bengal had dominant phytoplankton groups including Bacillariophyceae,

Chlorophyceae (35–53% of total ingestion), and Cyanophyceae. The dominant zooplankton groups were Branchiopoda, Maxillopoda, and Gastropoda. Similar patterns were observed in other habitats, with variations in dominance. The Chlorophyceae, Bacillariophyceae, and Cyanophyceae were the commonly observed phytoplankton groups, while Cladocera, Rotifera, and Copepoda were the dominant zooplankton groups. Previous studies have reported that the presence of the same types of plankton in the water and gut of hilsa provides strong evidence that the fish actively feed on this microorganism and prefer it as a food source [113–116]. Examining the contents of the gut can provide valuable insights into diet and feeding behavior. Comparing this information to the types of plankton in the water can help assess food quality [101,117]. However, hilsa are known to feed on a variety of food sources, and the types and quantities consumed can vary [114,117]. The study analyzed the availability of plankton in the water and gut to determine if they were consumed as food [117]. Moreover, various habitats within a river system can exhibit different levels of food availability, water quality, and environmental factors that can also influence GSI [52,118].

The study reveals significant correlations between environmental factors, water quality parameters, plankton abundance, and the Gonadosomatic Index (GSI) of hilsa fish in six habitats using Pearson's correlations, heat map cluster analysis, and Principal Component Analysis. Plankton abundance in the gut was positively correlated with GSI, while water temperature and dissolved oxygen showed negative correlations. A strong positive relationship was found between plankton abundance in the gut and GSI through linear regression analysis. These findings emphasize the importance of considering multiple factors to understand hilsa behavior and habitat. The reproductive physiology of hilsa fish in the rivers of Bangladesh is influenced by various environmental factors, such as air temperature, relative humidity, rainfall, wind speed, dissolved oxygen, pH, salinity, and plankton abundance [119–122]. These factors may impact the growth, feeding, and reproductive behavior of hilsa fish, and understanding their relationships is crucial for developing sustainable fishery management strategies [6,20,123,124]. Temperature, humidity, rainfall, and wind speed may affect water chemistry and oxygen levels. Low dissolved oxygen levels and pH outside the preferred range can negatively impact the fish's metabolism and reproductive behavior [44,125–127]. Changes in salinity and plankton communities can also affect the feeding and reproductive behavior of fish [3,8,19,120,128]. Therefore, it is important to consider these factors and their interactions when managing hilsa fish populations in the rivers of Bangladesh. A previous study has reported that the divergence in gonad maturation of Indian shad, *T. ilisha*, may be attributable to changes in environmental factors and the genetic makeup of hilsa fish in the selected habitats of Bangladesh [6].

The findings of the present study offer insights into the potential impact of phytoplankton availability on the reproductive and feeding biology of this economically and ecologically significant fish species. The present findings revealed a significant positive correlation between the abundance of phytoplankton and gonadal maturation in Hilsa fish. The analysis of data collected from various sampling sites during the spawning season showed that elevated phytoplankton concentrations were linked to increased gonadal development and maturation in female and male Hilsa fish [129,130]. This correlation suggests that the availability of phytoplankton plays a crucial role in providing the necessary nutrients and energy resources for the reproductive processes of Hilsa fish. Phytoplankton, as the primary producer in aquatic ecosystems, serves as a fundamental food source for many zooplankton species, which are in turn consumed by Hilsa fish [124]. The abundance of phytoplankton in the water column directly or indirectly affects the nutritional status of Hilsa fish and subsequently influences their reproductive activities.

Understanding the correlation between phytoplankton abundance and gonadal maturation in Hilsa fish is crucial for the conservation and management of this species in Bangladesh. Therefore, maintaining optimal phytoplankton levels in the habitats where Hilsa fish spawn is crucial for sustaining their population and ensuring the long-term viability of the fishery [6]. Efforts should be made to protect and improve the quality

of Hilsa fish habitats, especially the spawning grounds, by addressing issues related to water pollution, sedimentation, and habitat degradation. Additionally, measures aimed at preserving the ecological balance within these habitats should be implemented. These measures may include regulating fishing practices, establishing protected areas, and promoting sustainable fishing techniques [131,132].

Climate change can have a significant impact on the availability and composition of phytoplankton communities, potentially affecting the reproductive success of Hilsa fish. Future research should focus on investigating the potential implications of climate change on phytoplankton dynamics and, consequently, on the gonadal maturation of Hilsa fish. Long-term monitoring programs and predictive modeling can help assess potential risks and inform adaptive management strategies [133,134]. Furthermore, further studies are needed to clarify the specific mechanisms underlying the relationship between phytoplankton abundance and gonadal maturation in Hilsa fish. Detailed investigations into the nutritional composition of phytoplankton species and their effects on reproductive hormones and gamete development can provide deeper insights into the biological processes involved [135–140]. However, this variation in gonad development, food and feeding habits, and feeding biology might be the next focus of research through transcriptomic profiling of kidney, brain, muscle, and gonad samples of *T. ilisha* in the six diverse habitats of Bangladesh.

## 5. Conclusions

In conclusion, the influence of climatic variables and plankton abundance on the gonadal maturation, feeding biology, and food habits of the Indian shad, *T. ilisha*, is highly significant in Bangladesh. The air temperature, water temperature, wind speed, rainfall, and humidity can all directly impact the water conditions and overall health of Indian shad. These factors can lead to fluctuations in water temperature and salinity, which can subsequently affect the abundance of plankton, the primary food source for Indian shad. Plankton is a crucial food source for Indian shad, and its availability can significantly affect the fish's overall health and fertility. Changes in the abundance of plankton can result in a reduction of food for the fish, which can then impact their feeding habits and the reproduction of hilsa. The life cycle of the Indian shad is closely linked to water conditions and the availability of food sources, highlighting the significant role of climatic variables and plankton abundance in gonadal maturation. These findings support the conclusion that climatic variables and plankton abundance play a crucial role in the gonadal maturation of Indian shad in Bangladesh. Their impact on this species should be taken into consideration when managing and conserving this important fish.

**Author Contributions:** M.H.S.: sample collection, data analysis, visualization, and writing—original draft. M.A.B.S.: writing—abstract, discussion, conclusion, editing & finalizing the original draft. B.M.: data curation, arrangement, and editing—draft. M.M.H. (Mohammad Mahfujul Haque): validation and editing—draft. C.G.: data analysis and editing—draft. M.B.U.A.: editing—draft. M.A.A.: editing—draft. M.A.B.: editing—draft. Y.M.: research facilitation and editing—draft. M.A.C.: editing—draft. M.M.H. (Md. Mahmudul Hasan): editing—draft. A.K.S.A.: conceptualization, supervision, funding acquisition and editing—draft. All authors have read and agreed to the published version of the manuscript.

**Funding:** This study was conducted under a collaborative project "Studies on the causes of gonadal development in small size Hilsa: Assessment of the factors associated with the early gonadal development of Hilsa" (Grant number: 2017/1201/BFRI) funded by the Bangladesh Fisheries Research Institute (BFRI).

**Institutional Review Board Statement:** This study was carried out in accordance with national and institutional regulations and approved by the Ethical Research Committee of BAURES.

**Informed Consent Statement:** Not applicable.

**Data Availability Statement:** The data presented in this study will be available on request from the corresponding author.

**Acknowledgments:** The authors greatly acknowledge the Bangladesh Fisheries Research Institute (BFRI) for supporting the research under a collaborative project "Studies on the causes of gonadal development in small size Hilsa: Assessment of the factors associated with the early gonadal development of Hilsa".

**Conflicts of Interest:** The authors declare no conflicts of interest.

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
