# Peer review of "Multifaceted Linkages among Eco-Climatic Factors, Plankton Abundance, and Gonadal Maturation of Hilsa Shad, Tenualosa ilisha, Populations in Bangladesh"

_climate, doi:10.3390/cli12030040_

Round 1

Reviewer 1 Report

Comments and Suggestions for Authors

The present manuscript is devoted to the biological feeding habits of a unique hilsa shad fish, Tenualosa ilisha, collected from six different aquatic habitats in 19 districts of Bangladesh. In various types of habitat of this fish, the authors assessed the influence of climatic and environmental factors, composition and abundance of planktonic organisms living in the water on the reproductive characteristics of Tenualosa ilisha. An analysis of the food preferences of Tenualosa ilisha is demonstrated. The research was carried out by integrating all available data sets. Considering the economically and ecologically importance of Tenualosa ilisha, the results are of great practical meaning and will be useful to scientists, researchers and others involved in related research.

Some points require clarification.

1. Section 2.3.1, Lines 178-179. The authors used a plankton net, with a mesh size of 25 micrometers for phytoplankton and 90 micrometers for zooplankton. It is known that many representatives of phytoplankton and zooplankton can be much smaller in size. For example, many species of Chlorophyceae have cell sizes of 5-20 µm. In this case, we may get an incorrect percentage of phytoplankton groups in water samples.

What sizes were noted for planktonic organisms in water, i.e. what size fractions did they belong to? What sizes were the plankton species in the gut? Is there any selectivity of Tenualosa ilisha to the size parameters of its food?

2. Figure 1, Figure 2. The figures show the Percentage composition of Plankton abundance in water and in the gut. It is not clear what the authors mean by the term “abundance” - is it the number of counted organisms (cells) from different groups of plankton or is it an indicator of species diversity, i.e. number of genera identified in groups. If the authors mean species richness, then it is more correct to write the term “diversity” – “Percentage taxonomical diversity of different plankton groups in water”. The signatures in these figures are not clear.

3. Section 3.4. Lines 322-335. The authors give coefficient values either as percentages or numbers. It needs to be done the same way.

4. Section 3.4. Line 350. Was there a procedure for determining the optimal number of clusters?

5. Figure 6. This Figure must be accompanied either by a table or in the text by specifying which factor explains how much variance. It is important to understand the contribution of each factor to the overall model. The color coding needs to be changed. Or each point on the raft must have a signature. The reader should understand which samples are grouped, where and how. Based on this, by the way, we can talk about the influence of factors on the plankton of these particular groups of samples.

6. Figure 7. These Figures must be positioned vertically and enlarged.

Author Response

Response to reviewer comments-1

Open Review

Quality of English Language

(x) I am not qualified to assess the quality of English in this paper
( ) English very difficult to understand/incomprehensible
( ) Extensive editing of English language required
( ) Moderate editing of English language required
( ) Minor editing of English language required
( ) English language fine. No issues detected  

Yes

Can be improved

Must be improved

Not applicable

Does the introduction provide sufficient background and include all relevant references?

(x)

( )

( )

( )

Are all the cited references relevant to the research?

(x)

( )

( )

( )

Is the research design appropriate?

( )

(x)

( )

( )

Are the methods adequately described?

( )

(x)

( )

( )

Are the results clearly presented?

( )

(x)

( )

( )

Are the conclusions supported by the results?

( )

(x)

( )

( )

Comments and Suggestions for Authors

The present manuscript is devoted to the biological feeding habits of a unique hilsa shad fish, Tenualosa ilisha, collected from six different aquatic habitats in 19 districts of Bangladesh. In various types of habitat of this fish, the authors assessed the influence of climatic and environmental factors, composition and abundance of planktonic organisms living in the water on the reproductive characteristics of Tenualosa ilisha. An analysis of the food preferences of Tenualosa ilisha is demonstrated. The research was carried out by integrating all available data sets. Considering the economically and ecologically importance of Tenualosa ilisha, the results are of great practical meaning and will be useful to scientists, researchers and others involved in related research.

Response to the reviewer comments: Thank you for your appreciation and acknowledgement of the significance and potential of our study. We have carefully reviewed the whole manuscript with thoughtful consideration and made revisions based on the reviewer's comments in order to improve it to the desired standard.

Some points require clarification.

  1. Section 2.3.1, Lines 178-179. The authors used a plankton net, with a mesh size of 25 micrometers for phytoplankton and 90 micrometers for zooplankton. It is known that many representatives of phytoplankton and zooplankton can be much smaller in size. For example, many species of Chlorophyceae have cell sizes of 5-20 µm. In this case, we may get an incorrect percentage of phytoplankton groups in water samples.

What sizes were noted for planktonic organisms in water, i.e. what size fractions did they belong to? What sizes were the plankton species in the gut? Is there any selectivity of Tenualosa ilisha to the size parameters of its food?

Response to the reviewer comments: Thank you for your valuable comments and suggestions. We noticed that various studies have shown the use of different mesh sizes for plankton collection in various aspects, but we purposely chose plankton nets with a 10-micrometer mesh size for phytoplankton and a 90-micrometer mesh size for zooplankton. It was a typographical mistake. Moreover, we did not consider the measurement of plankton size for water or gut content. Our aim was to collect plankton in numbers for both phytoplankton and zooplankton, rather than focusing on size. This is an important aspect that should be addressed in future experiments, where selectivity to size parameters of food can be investigated.

  1. Figure 1, Figure 2. The figures show the Percentage composition of Plankton abundance in water and in the gut. It is not clear what the authors mean by the term “abundance” - is it the number of counted organisms (cells) from different groups of plankton or is it an indicator of species diversity, i.e. number of genera identified in groups. If the authors mean species richness, then it is more correct to write the term “diversity” – “Percentage taxonomical diversity of different plankton groups in water”. The signatures in these figures are not clear.

Response to the reviewer comments: Thank you for your valuable comments. We mean plankton abundance in both water and gut in numerical terms for all cases. To make this clearer, we have revised it to read as "plankton abundance in numbers in water/gut" in the revised manuscript. We would appreciate any further advice or suggestions for improvement.

  1. Section 3.4. Lines 322-335. The authors give coefficient values either as percentages or numbers. It needs to be done the same way.

Response to the reviewer comments: Thank you for your valuable advice. The coefficient values are made same styles in the revised manuscript as track changes.   

  1. Section 3.4. Line 350. Was there a procedure for determining the optimal number of clusters?

Response to the reviewer comments: Thank you for your comment. A procedure was followed to determine the optimal number of clusters. To determine the optimal number of clusters, we utilized hierarchical clustering with the complete linkage method, using Euclidean distance as the metric for similarity assessment.

  1. Figure 6. This Figure must be accompanied either by a table or in the text by specifying which factor explains how much variance. It is important to understand the contribution of each factor to the overall model. The color coding needs to be changed. Or each point on the raft must have a signature. The reader should understand which samples are grouped, where and how. Based on this, by the way, we can talk about the influence of factors on the plankton of these particular groups of samples.

Response to the reviewer comments: Thank you for your advice. The table is included to demonstrate the variance indicated by the different components. The color coding clearly indicates the contribution of each parameter to the overall relationship. We believe that readers will easily understand and see the significance of each parameter.

  1. Figure 7. These Figures must be positioned vertically and enlarged.

Response to the reviewer's comments: The figure has been positioned vertically in the manuscript, as advised. Thank you for this important suggestion.

Reviewer 2 Report

Comments and Suggestions for Authors

Abstract:

·        Key points of Methodology are missing in the Abstract. Authors have jumped directly into Results after Introduction.

·        L23: Plankton or shad? Not clear.

·        L36-39: Two sentences repeating similar kinds of results.

·        L39: Better to define abbreviation “GSI” at first appearance.

Methodology:

·        Table 1: Needs formatting to separately identify site names

·        L 147: “Kali River are upstream habitats, while the Meghna River, Padma  River, Tetulia River, and Bay of Bengal are downstream habitats.” Relative to what?

·        Better to include a site map.

·        L 171: “… meet the growing demand for important aspects and indicators.” On what?

·        L 197: number of species or individuals/ Liter?

·        L231-235: repeating information.

Results:

·        L300-303: better if you can provide proportions of taxa. E.g. …. 14/ (total of that taxon outside) of Chlorophyceae …

·        L 304-305: “The overall electivity index results showed that hilsa preferred phytoplankton over zooplankton.” This is too speculative since exact zero value is rare and resulted values are close to zero.

·        L 322-324: “According to the correlation matrix (Figure 3), significant correlations were found  among the climatic variables, water quality parameters, plankton abundance in water and gut, and the gonadosomatic index of T. ilisha.”

It is not clear which plankton abundance has been used here since you claimed identification up to genus level.

According to Methodology, water quality and GSI data were adopted from the previous study, which has been published in 2021 and plankton data have been collected later as I understood. If that is the case, checking for correlations between datasets collected at 2 different periods of time is not correct.

·        L 349-357: mention the similarities between clusters.

·        L 361-387: Those PCA analyses need a revision. If you consider plankton abundances and GSI as dependent variables of environmental variability, they belong to 2 data matrices and are not suitable for being used together as one matrix and run a PCA. A CCA would be a better option.

Comments on the Quality of English Language

English is in acceptable level but there are instances of misusage of "the" all over the document.

Author Response

Response to reviewer comments-2

Open Review

Quality of English Language

( ) I am not qualified to assess the quality of English in this paper
( ) English very difficult to understand/incomprehensible
( ) Extensive editing of English language required
( ) Moderate editing of English language required
(x) Minor editing of English language required
( ) English language fine. No issues detected

Yes

Can be improved

Must be improved

Not applicable

Does the introduction provide sufficient background and include all relevant references?

(x)

( )

( )

( )

Are all the cited references relevant to the research?

(x)

( )

( )

( )

Is the research design appropriate?

( )

(x)

( )

( )

Are the methods adequately described?

( )

( )

(x)

( )

Are the results clearly presented?

( )

(x)

( )

( )

Are the conclusions supported by the results?

( )

(x)

( )

( )

Response to the reviewer comments: Thank you for your appreciation and acknowledgement of the significance and potential of our study. We have carefully reviewed the manuscript with thoughtful consideration and made revisions based on the reviewer's comments in order to improve it to the desired standard.

Comments and Suggestions for Authors

Abstract:

Key points of Methodology are missing in the Abstract. Authors have jumped directly into Results after Introduction.

Response to the reviewer comments: Thank you for your valuable suggestions. We have looked through the abstract part again sincerely and included short methodology in the revised manuscript.        

L23: Plankton or shad? Not clear.

Response to the reviewer comments: Thank you for your comment. In our experiment, plankton refers to the composition in numbers found in both water and the gut. For Hilsa shad, it specifically refers to the gonadosomatic index during the study period.

L36-39: Two sentences repeating similar kinds of results.

Response to the reviewer comments: It has been corrected in the revised manuscript.

L39: Better to define abbreviation “GSI” at first appearance.

Response to the reviewer comments: It has been corrected in the revised manuscript.

Methodology:

Table 1: Needs formatting to separately identify site names

Response to the reviewer comments: Thank you for your comment. We have collected representative samples from three different GPS points along the river, Haor, and Bay, all of which were relatively close to each other. Therefore, it is difficult to give site names in the water locations. However, we have revised the table structure to make it simpler and easier to understand, and it is now included in the revised manuscript.

L 147: “Kali River are upstream habitats, while the Meghna River, Padma River, Tetulia River, and Bay of Bengal are downstream habitats.” Relative to what?

Better to include a site map.

Response to the reviewer comments: Thanks for valuable advice. Bangladesh map with study area has been included in the revised manuscript.

L 171: “… meet the growing demand for important aspects and indicators.” On what?

Response to the reviewer comments:  It has been update in the revised manuscript. Thanks for informative comments.      

L 197: number of species or individuals/ Liter?

Response to the reviewer comments: The ideal number of species per liter should be good for interpreting data. Thanks again.

L231-235: repeating information.

Response to the reviewer comments: The repetition has been eliminated and revised. Thank you again.

Results:

L300-303: better if you can provide proportions of taxa. E.g. …. 14/ (total of that taxon outside) of Chlorophyceae …

Response to the reviewer comments:  Thank you for this comment. Plankton is proportionately mentioned in the same section, starting from line 290. Hopefully, this makes more sense now.

L 304-305: “The overall electivity index results showed that hilsa preferred phytoplankton over zooplankton.” This is too speculative since exact zero value is rare and resulted values are close to zero.

Response to the reviewer comments:  Thank you for your observations and comments. We highlighted our statement emphasizes the concept that positive values signify preference and a higher positive value indicates a stronger preference. On the other hand, negative values indicate avoidance and a higher negative value indicates a stronger aversion. However, instead of discussing the value range, our focus is on understanding the meanings of positive and negative values.

L 322-324: “According to the correlation matrix (Figure 3), significant correlations were found among the climatic variables, water quality parameters, plankton abundance in water and gut, and the gonadosomatic index of T. ilisha.”

It is not clear which plankton abundance has been used here since you claimed identification up to genus level.

Response to the reviewer comments: Thank you for your comment. We considered values for climatic variables, water quality parameters, plankton abundance in water, and gut for the correlation matrix on an annual average basis. This has been revised in the respective line.

According to Methodology, water quality and GSI data were adopted from the previous study, which has been published in 2021 and plankton data have been collected later as I understood. If that is the case, checking for correlations between datasets collected at 2 different periods of time is not correct.

Response to the reviewer comments: Thank you for this question. All of the data were collected during the same study period under the same project. The article was previously published based on some of the data. We believe the matter is now clear to you.

L 349-357: mention the similarities between clusters.  

Response to the reviewer comments: In the gut source cluster, small numbers of phytoplankton groups (cyanophyceae, Dinophyceae, and Xanthophyceae) and large numbers of zooplankton groups (Protozoan, Hydrozoa, Polychaeta, Cladocera, Gastropod, and Copepoda) were less dominant. Conversely, these groups showed high dominance in the water source cluster.

L 361-387: Those PCA analyses need a revision. If you consider plankton abundances and GSI as dependent variables of environmental variability, they belong to 2 data matrices and are not suitable for being used together as one matrix and run a PCA. A CCA would be a better option.

Response to the reviewer comments: Thank you for raising a great question regarding the use of PCA and CCA approaches. The connection between eco-climatic factors, plankton abundance, and Gonadosomatic index is a complex phenomenon. We took a closer look at this issue when conducting our analysis. Ideally, PCA establishes a proportional connection between all parameters. However, if we consider plankton abundance and GSI as dependent variables for CCA, the impact of plankton on GSI could potentially be overlooked. Thus, we believe that PCA is the most comprehensive approach for analyzing the influence among variables.

Comments on the Quality of English Language     

English is in acceptable level but there are instances of misusage of "the" all over the document.

Response to the reviewer comments: Thank you for your helpful suggestion. We have carefully reviewed the entire manuscript for English language errors and have made necessary improvements through track changes, as required in the manuscript. All the changes are also highlighted as yellow shading.

Round 2

Reviewer 2 Report

Comments and Suggestions for Authors

Please check the red font

Comments on the Quality of English Language

Please re-arrange the highlighted sentence in the attachment to increase readability.

Author Response

Date: 19.2.24

Response to reviewer comments 2

According to the comments from reviewer 2, we have thoroughly checked the entire manuscript, including the English language editing. We have marked track changes with yellow shading for easy identification. The only yellow highlighted sections are previous corrections made according to the comments of previous reviewers (1 & 2). We are grateful for your valuable feedback and advice, which have greatly improved the manuscript.
